# IGD: Token Decisiveness Modeling via Information Gain in LLMs for Personalized Recommendation

**Zijie Lin**[1]**, Yang Zhang**[1†]**, Xiaoyan Zhao**[2]**, Fengbin Zhu**[1†]**, Fuli Feng**[3]**, Tat-Seng Chua**[1]

[1]National University of Singapore     [2]The Chinese University of Hong Kong
[3]University of Science and Technology of China
zijie.lin@u.nus.edu, zyang1580@gmail.com
xzhao@se.cuhk.edu.hk, zhfengbin@gmail.com
fulifeng93@gmail.com, dcscts@nus.edu.sg
[†]Corresponding author

## Abstract

Large Language Models (LLMs) have shown strong potential for recommendation by framing item prediction as a token-by-token language generation task. However, existing methods treat all item tokens equally, simply pursuing likelihood maximization during both optimization and decoding. This overlooks crucial token-level differences in decisiveness—many tokens contribute little to item discrimination yet can dominate optimization or decoding. To quantify token decisiveness, we propose a novel perspective that models item generation as a decision process, measuring token decisiveness by the Information Gain (IG) each token provides in reducing uncertainty about the generated item. Our empirical analysis reveals that most tokens have low IG but often correspond to high logits, disproportionately influencing training loss and decoding, which may impair model performance. Building on these insights, we introduce an Information Gain-based Decisiveness-aware Token handling (IGD) strategy that integrates token decisiveness into both tuning and decoding. Specifically, IGD downweights low-IG tokens during tuning and rebalances decoding to emphasize tokens with high IG. In this way, IGD moves beyond pure likelihood maximization, effectively prioritizing high-decisiveness tokens. Extensive experiments on four benchmark datasets with two LLM bbones demonstrate that IGD consistently improves recommendation accuracy, achieving significant gains on widely used ranking metrics compared to strong baselines. Our codes are available at https://github.com/ZJLin2oo1/IGD.

## 1 Introduction

Recommendation systems [1, 2, 3] play a crucial role in helping users discover relevant and personalized content. With recent advances in Large Language Models (LLMs) [4, 5, 6], there is growing interest in leveraging LLMs' strong language understanding and reasoning capabilities [7, 8, 9] for recommendation tasks [10, 11, 12, 13, 14, 15], giving rise to a new paradigm known as LLM4Rec [16, 17, 18]. In this paradigm, recommendation is typically formulated as a natural language problem: user history and task context are encoded into a prompt, based on which the LLM is tuned to generate the top-$K$ recommended items via autoregressive token decoding [19]. This approach has demonstrated strong performance in capturing user intent and generating personalized outputs [20, 21, 22].

Despite its promise, we argue that the existing token handling strategy in LLM4Rec does not fully align with the item generation process in recommendation. The current approach is primarily likelihood-driven, treating each item token equally and simply focusing on: 1) optimizing token

likelihood during fine-tuning for data fitting [23, 24, 21], and 2) maximizing token likelihood during inference for generation [19, 20]. However, not all tokens are equally important. Some tokens are more decisive in defining the item, while others serve grammatical or filler functions with low decisiveness. Low-decisiveness tokens do not reduce uncertainty in item generation, making their focus less meaningful. Moreover, low-decisiveness tokens may have high logits—such as "ghost tokens" defined in [19], which have generation probabilities close to 1 for a given prefix. These tokens can dominate likelihood-maximizing decoding, introducing bias toward items with more such tokens [19]. To improve both tuning and decoding, it is crucial to quantify and incorporate token decisiveness, rather than relying solely on likelihood optimization.

To measure token decisiveness, we introduce a novel perspective that frames token-by-token item generation in LLM4Rec as a decision process. In this framework, the uncertainty of the recommendation outcome at each generation step is quantified by the entropy [25] of the item distribution conditioned on the tokens generated so far. The decisiveness of a token is defined as the reduction in this uncertainty when selecting the token, *i.e.*, its Information Gain (IG) [26]. Based on this definition, we observe that: 1) in the studied real-world datasets, over 50% of item tokens exhibit zero IG; and 2) under the existing token strategy, LLM4Rec models may be misled by these zero-IG tokens. Specifically, as shown in Figure 5, models tend to over-optimize zero-IG tokens while under-emphasizing non-zero-IG tokens during tuning. Furthermore, as shown in Figure 4, low IG tokens, especially zero-IG tokens, often correspond to high logits, which can bias the likelihood-maximizing decoding process toward items containing more such tokens.

To incorporate token decisiveness into tuning and decoding, we propose an *Information Gain-based Decisiveness-aware Token Handing* (IGD) strategy that goes beyond simply optimizing/maximizing token likelihood. During tuning, IGD downweights zero-IG tokens to prioritize informative (non-zero-IG) tokens that aid in item discrimination, enabling more effective learning. At decoding, IGD increases the influence of high IG tokens along the autoregressive path by adjusting focus toward their logits, rather than blindly following the likelihood-maximizing principle, thereby reducing bias toward items dominated by high-logit but low-decisiveness tokens. In this way, IGD reshapes token importance by incorporating token decisiveness, leading to improved recommendation accuracy. We evaluate IGD on four benchmark datasets using two LLM backbones. Results demonstrate that IGD consistently enhances recommendation performance, with average gains of 18.89% in HR@10 and 20.15% in NDCG@10 over strong baselines.

The main contributions are: (1) We emphasize the importance of quantifying and incorporating token decisiveness in both tuning and decoding, and propose framing the item generation process as a decision process, defining token decisiveness based on information gain. (2) We introduce IGD, a simple yet effective token handling strategy that leverages token-level information gain to guide both tuning and decoding, going beyond mere likelihood optimization/maximization to prioritize high-decisiveness tokens, thereby enhancing recommendation performance. (3) We conduct extensive experiments on four benchmark datasets using two LLM backbones. The results show that IGD consistently improves recommendation performance, achieving significant gains in HR@10 and NDCG@10 over strong baselines.

## 2 Preliminary

This section introduces the typical tuning and decoding approaches in LLM4Rec.

### 2.1 Tuning

To enable next-item prediction using LLMs, supervised fine-tuning (SFT) is usually applied to teach LLMs to map items to token sequences. Given an input prompt $x$ transformed from user history and task description, and a list of target item tokens $y = (y_1, \ldots, y_m)$, the model is trained to minimize the token-level cross-entropy loss:

$$\mathcal{L} = \sum_{t=1}^{m} \ell(f_\theta(x, y_{<t}); y_t), \tag{1}$$

where $f_\theta$ denotes the LLM with parameters $\theta$, and $\ell$ is the cross-entropy between the predicted token distribution and the ground-truth token $y_t$ at position $t$.

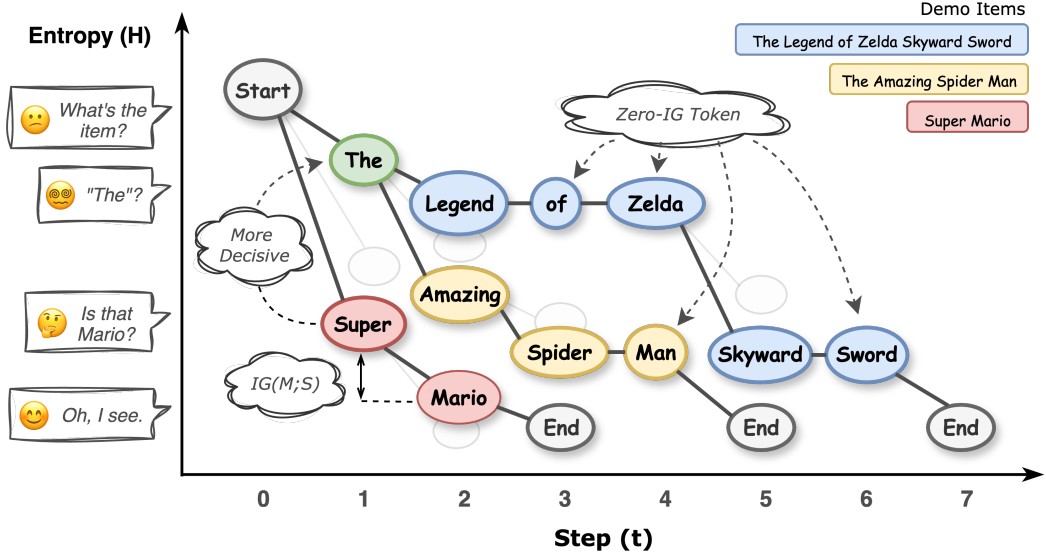

Figure 1: Illustration of LLM4Rec autoregressive token generation as a sequential decision process. As tokens are generated, the entropy of the remaining sequence gradually decreases. The information gain (IG) quantifies this reduction, e.g., $IG(M; S)$ measures the IG of token "Mario" given prefix "Super". Tokens shared across many items (e.g., "The") exhibit lower decisiveness with lower IG, while more decisive tokens (e.g., "Super") lead to larger IG. Tokens with IG=0—such as "of", "Zelda", "Sword", and "Man"—are referred to as zero-IG tokens.

## 2.2 Decoding

**Autoregressive Decoding.** At inference time, the model generates item sequences autoregressively. The conditional probability of a full sequence $y$ given $x$ is factorized as:

$$p(y|x) = \prod_{t=1}^{m} p(y_t|x, y_{<t}) \tag{2}$$

This formulation enables step-by-step generation but highly relies on local token-level probabilities.

**Beam Search Decoding.** During inference, existing methods commonly use beam search to generate multiple item candidates simultaneously. At each decoding step $t$, the model expands each partial sequence $y_{\leq t-1}$ by considering the top-ranked token candidates, and updates the cumulative score using:

$$S(y_{\leq t}) = S(y_{\leq t-1}) + \log p(y_t|x, y_{<t}), \tag{3}$$

where $S(y_{\leq t})$ denotes the log-probability of the sequence prefix $y_{\leq t}$, and $p(y_t|x, y_{<t})$ is the conditional probability of the token in step $t$.

In standard natural language generation tasks, a length penalty is often introduced to avoid overly long outputs. However, recent work [19] reveals that applying such penalties in recommendation scenarios tends to favor longer item sequences, introducing bias into the final selection. Therefore, we follow [19] and set the length penalty to zero, keeping the score computation purely based on accumulated log-probabilities as shown above.

# 3 Token Decisiveness Modeling

## 3.1 Token Decisiveness Measurement

We model the token generation process in LLM4Rec as a sequential decision-making procedure, where each autoregressive decoding step progressively refines the target item from the full item space. Let $\mathcal{I}$ denote the full item collection, and $\mathcal{I}^{y_{\leq t}} \subseteq \mathcal{I}$ represent the set of candidate items consistent with the generated token prefix $y_{\leq t}$ at step $t$. Following information theory [27], we quantify the uncertainty of the recommendation at step $t$ using Shannon entropy [25] over the candidate item distribution:

$$H(y_{\leq t}) = - \sum_{\mathcal{I}_i \in \mathcal{I}^{y_{\leq t}}} p_r(\mathcal{I}_i) \log p_r(\mathcal{I}_i), \tag{4}$$

where $p_r(\mathcal{I}_i)$ denotes the empirical prior probability of item $\mathcal{I}_i$ estimated from the training data. The entropy is computed over the candidate set $\mathcal{I}^{y_{\leq t}}$ compatible with the current token prefix.

At each step, a new token $y_t$ reduces the candidate space. We measure the decisiveness of $y_t$ using its **Information Gain (IG)** —the reduction in uncertainty it induces:

$$\mathrm{IG}(y_t; y_{<t}) = H(y_{\leq t-1}) - H(y_{\leq t}) \tag{5}$$

This formulation quantifies how much $y_t$ contributes to identifying the target item, where a higher IG indicates greater decisiveness.

## 3.2 Statistical Analysis on Token Decisiveness

Table 1: Dataset Statistics and zero-IG Token Proportion

| Dataset | Items | Train | Valid | Test | Tokens | zero-IG Tokens (%) |
|---------|-------|-------|-------|------|--------|--------------------|
| CDs | 14,239 | 148,685 | 18,586 | 18,587 | 805,786 | 450,960 (55.96%) |
| Games | 11,037 | 201,613 | 25,202 | 25,203 | 2,128,430 | 1,292,171 (60.71%) |
| Toys | 11,252 | 112,755 | 14,095 | 14,096 | 1,530,370 | 1,098,070 (71.75%) |
| Books | 41,722 | 682,998 | 85,376 | 85,376 | 7,183,839 | 5,241,997 (72.97%) |

We conduct a statistical analysis of the proposed IG metrics on four public datasets, summarized in Table 1. First, each item title is tokenized into a sequence of tokens using the Qwen2.5 tokenizer [28]. Then, we construct a token prefix tree and compute, for each prefix, its corresponding entropy as well as the IG for each token. All statistics are computed exclusively on the training set. From Table 1, we observe that zero-IG tokens—tokens that yield no reduction in entropy are predominant, constituting 55.96% to 72.97% of all tokens across datasets.

## 3.3 Token-level Biases

To investigate how LLMs interact with token decisiveness, we analyze tuning dynamics and decoding paths using the D3 [19] method with the Qwen2.5-1.5B model [28]. In model tuning, we respectively monitor the tuning loss for zero-IG tokens and non-zero-IG tokens across the training set; In model decoding, we gather both ground-truth items and top-10 predicted items from the test set, compute the average entropy of the prefix at each decoding step, and compare the average entropies between predicted and ground-truth prefixes.

We observe the following biases:

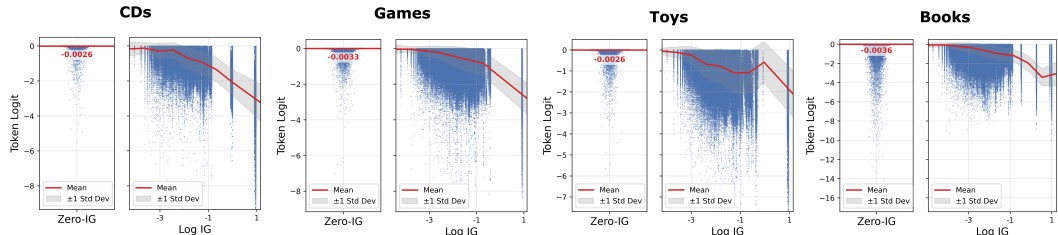

Figure 4: Relationship between IG values and logits of tokens in decoding. For each dataset, the left subfigure shows that zero-IG tokens are associated with extremely high logits (close to 0). The right subfigure illustrates a negative correlation between IG values and logit magnitudes for non-zero-IG tokens.

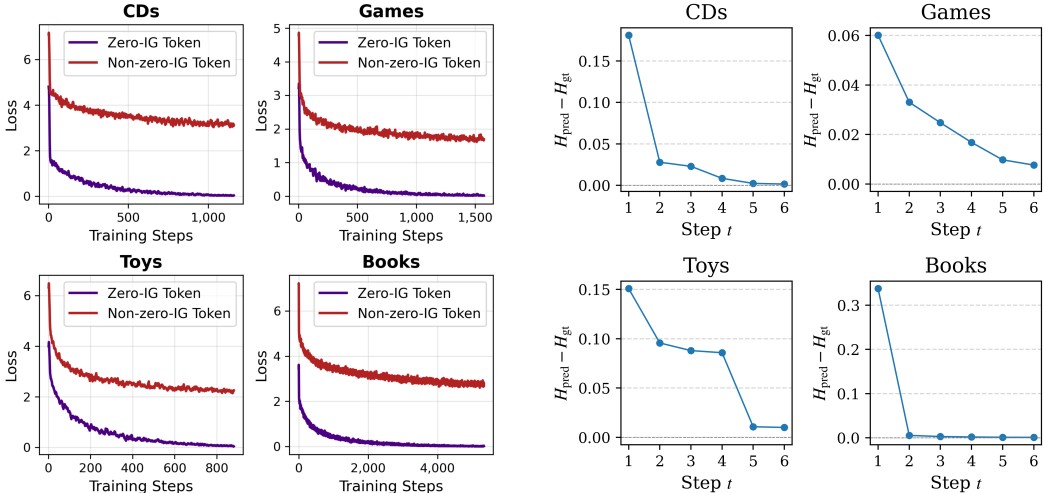

Figure 2: Loss comparison between zero-IG and non-zero-IG tokens in model tuning (epoch 1)

Figure 3: Entropy difference in decoding: model prediction vs. ground-truth

**Tuning Bias:** During model tuning, models tend to over-optimize zero-IG tokens while under-emphasizing non-zero-IG tokens. As shown in Figure 2, the model rapidly minimizes the loss on zero-IG tokens, while the loss on non-zero-IG tokens remains relatively high. This imbalance leads to a biased learned distribution, causing the LLM to favor less decisive tokens. Therefore, a more effective training approach that emphasizes decisive tokens is necessary.

**Decoding Bias:** As shown in Figure 4, LLMs tend to assign higher logits to low IG tokens, particularly those with zero IG. Since beam search selects top-$k$ candidates solely based on token-level logit scores, these low IG tokens are favored during decoding. As a result, the generated item prefixes exhibit higher average entropy than the ground-truth prefixes ( Figure 3 ), indicating a shift toward less informative predictions. This reveals a decoding bias, where the model systematically prefers less decisive tokens due to the likelihood-based decoding objective.

## 4 Information Gain-based Decisiveness-aware Token Handing (IGD)

To mitigate the token-level bias during both tuning and decoding phases, we propose a two-stage method, **I**nformation **G**ain-based **D**ecisiveness-aware Token Handling (**IGD**). Specifically, we leverage IG to quantify token decisiveness and adjust training dynamics and inference process accordingly.

**IGD-Tuning.** To mitigate learning bias towards non-decisive tokens, we introduce a token-level reweighting scheme into the loss function when tuning LLMs. The revised objective is:

$$\mathcal{L}_{\text{IGD}} = \frac{1}{\Omega} \sum_{t=1}^{|y|} w_t \cdot \ell(f_\theta(x, y_{<t}), y_t), \qquad (6)$$

where $w_t$ is a weight assigned to token $y_t$, $\Omega$ is the sum of $w_t$ across all predicted tokens. $w_t$ is defined as:

$$w_t = \begin{cases} \beta, & \text{if IG}(y_t; y_{<t}) = 0 \\ 1, & \text{if IG}(y_t; y_{<t}) > 0 \end{cases} \tag{7}$$

Here, $\beta \in [0, 1]$ is a hyperparameter that controls the penalty on non-decisive tokens, thereby reducing the learning focus on tokens with zero IG. Setting $\beta = 1$ recovers the standard cross-entropy loss.

**IGD-Decoding.** To address decoding bias that favors generic or homogeneous tokens, we modify the standard beam search scoring function to promote decisive tokens. The revised score at step $t$ is computed as:

$$S(y_{\leq t}) = S(y_{\leq t-1}) + w_d \cdot \log p(y_t | x, y_{<t}), \tag{8}$$

with the reweighting factor $w_d$ computed as:

$$w_d = 1 - \alpha \cdot \widetilde{\text{IG}}(y_t) \tag{9}$$

Here, $\widetilde{\text{IG}}(y_t)$ denotes the max-min normalized IG of token $y_t$ within the current beam step, scaled to $[0, 1]$ across all candidates. If all candidates have zero-IG, their normalized values are set to zero. The hyperparameter $\alpha \in [0, 1]$ controls the strength of decisiveness calibration. When $\alpha = 1$, the method falls back to standard beam search scoring.

## 5 Experiments

In this section, we aim to address the following research questions (RQs): **RQ1:** Does IGD improve the recommendation accuracy of LLM4Rec? **RQ2:** How does each stage of IGD contribute to performance improvements? **RQ3:** How does IGD influence the tuning and decoding to enhance performance? **RQ4:** Is IGD effective across LLMs of different scales and tokenization schemes?

### 5.1 Experimental Setup

**Datasets.** We evaluate IGD on four publicly available Amazon review datasets [29]: *CDs*, *Games*, *Toys*, and *Books*, covering data from May 1996 to October 2018. Dataset statistics are summarized in Table 1. Following the preprocessing procedure in the D3 paper [19], we truncate the data based on timestamps and filter out infrequent users and items, ensuring that each user and each item has at least 5 interactions.

**Compared Methods.** For standard recommendation settings, we compare our method against: (1) two representative sequential recommendation models—**GRU4Rec**[30], **SAS-Rec**[31] and **LRURec**[32]; and (2) two state-of-the-art (SOTA) LLM-based recommendation approaches—**BIGRec**[20] and **D3**[19]. Our IGD strategy can be integrated into both BIGRec and D3 for fair comparison. To further evaluate the effectiveness of IGD as a token handling strategy, we compare it with two token handing baselines that can also be seamlessly integrated into LLM4Rec frameworks: 1) **Position Normalization (Pos)**, which assigns higher weight to earlier item tokens to mitigate position bias; and 2) **Causal Fine-tuning (CFT)** [33], which builds upon Pos by introducing an additional causal loss term to enhance the modeling of causal effects at the token level. See Appendix A for detailed descriptions of all baselines. By default, we implement all LLM-based methods using Qwen2.5-1.5B [28]. More implementation details, including hyperparameter tuning settings, can be found in Appendix B.

**Evaluation Metrics.** To evaluate the model's top-K recommendation performance, we adopt two widely-used metrics: Hit Ratio (HR@K) and Normalized Discounted Cumulative Gain (NDCG@K) [34]. Both metrics are computed under the all-ranking protocol [35], where the model ranks all candidate items for each user. In our experiments, we report results for $K = 5$ and $K = 10$.

Table 2: Recommendation performance of the compared methods evaluated on four benchmark datasets. H@K and N@K denote HR@K and NDCG@K, respectively. *Improvement* indicates the relative performance gain of IGD over the corresponding LLM4Rec backbone without any token reweighting. The best results are bold.

| Methods | CDs | | | | Games | | | |
|---|---|---|---|---|---|---|---|---|
| | N@5 | N@10 | H@5 | H@10 | N@5 | N@10 | H@5 | H@10 |
| GRU4Rec | 0.0248 | 0.0288 | 0.0342 | 0.0467 | 0.0169 | 0.0221 | 0.0261 | 0.0423 |
| SASRec | 0.0477 | 0.0535 | 0.0647 | 0.0824 | 0.0237 | 0.0290 | 0.0338 | 0.0502 |
| LRURec | 0.0540 | 0.0586 | 0.0680 | 0.0824 | 0.0298 | 0.0363 | 0.0421 | 0.0621 |
| BIGRec | 0.0502 | 0.0553 | 0.0623 | 0.0782 | 0.0317 | 0.0381 | 0.0430 | 0.0631 |
| +Pos | 0.0511 | 0.0566 | 0.0632 | 0.0802 | 0.0319 | 0.0396 | 0.0423 | 0.0665 |
| +CFT | 0.0509 | 0.0566 | 0.0631 | 0.0810 | 0.0349 | 0.0414 | 0.0482 | 0.0686 |
| **+IGD** | **0.0540** | **0.0593** | **0.0669** | **0.0833** | **0.0423** | **0.0507** | **0.0576** | **0.0833** |
| *Improvement* | *+7.78%* | *+7.82%* | *+9.33%* | *+9.04%* | *+33.4%* | *+33.1%* | *+34.0%* | *+32.0%* |
| D3 | 0.0716 | 0.0767 | 0.0882 | 0.1040 | 0.0415 | 0.0477 | 0.0581 | 0.0773 |
| +Pos | 0.0729 | 0.0779 | 0.0902 | 0.1053 | 0.0429 | 0.0489 | 0.0581 | 0.0767 |
| +CFT | 0.0736 | 0.0786 | 0.0917 | 0.1069 | 0.0437 | 0.0499 | 0.0613 | 0.0806 |
| **+IGD** | **0.0748** | **0.0801** | **0.0929** | **0.1092** | **0.0518** | **0.0598** | **0.0705** | **0.0946** |
| *Improvement* | *+4.47%* | *+4.43%* | *+5.33%* | *+5.00%* | *+25.6%* | *+29.2%* | *+26.7%* | *+22.7%* |

| Methods | Toys | | | | Books | | | |
|---|---|---|---|---|---|---|---|---|
| | N@5 | N@10 | H@5 | H@10 | N@5 | N@10 | H@5 | H@10 |
| GRU4Rec | 0.0200 | 0.0238 | 0.0275 | 0.0392 | 0.0060 | 0.0078 | 0.0094 | 0.0149 |
| SASRec | 0.0356 | 0.0398 | 0.0473 | 0.0745 | 0.0097 | 0.0123 | 0.0146 | 0.0226 |
| LRURec | 0.0358 | 0.0404 | 0.0463 | 0.0608 | 0.0257 | 0.0277 | 0.0319 | 0.0383 |
| BIGRec | 0.0553 | 0.0623 | 0.0736 | 0.0951 | 0.0190 | 0.0211 | 0.0245 | 0.0309 |
| +Pos | 0.0561 | 0.0631 | 0.0741 | 0.0958 | 0.0197 | 0.0218 | 0.0255 | 0.0319 |
| +CFT | 0.0561 | 0.0630 | 0.0746 | 0.0961 | 0.0195 | 0.0218 | 0.0250 | 0.0321 |
| **+IGD** | **0.0577** | **0.0656** | **0.0771** | **0.1014** | **0.0267** | **0.0294** | **0.0334** | **0.0419** |
| *Improvement* | *+4.34%* | *+5.30%* | *+4.76%* | *+6.62%* | *+41.3%* | *+40.0%* | *+36.9%* | *+36.0%* |
| D3 | 0.0634 | 0.0698 | 0.0833 | 0.1029 | 0.0212 | 0.0228 | 0.0266 | 0.0315 |
| +Pos | 0.0644 | 0.0702 | 0.0850 | 0.1029 | 0.0221 | 0.0237 | 0.0275 | 0.0324 |
| +CFT | 0.0640 | 0.0704 | 0.0840 | 0.1036 | 0.0219 | 0.0236 | 0.0275 | 0.0327 |
| **+IGD** | **0.0658** | **0.0726** | **0.0868** | **0.1082** | **0.0291** | **0.0313** | **0.0356** | **0.0424** |
| *Improvement* | *+3.79%* | *+4.01%* | *+4.20%* | *+5.15%* | *+37.3%* | *+37.3%* | *+33.8%* | *+34.6%* |

## 5.2 Main Results (RQ1)

In this section, we evaluate whether the proposed IGD method improves overall recommendation performance. Table 2 summarizes the performance of all compared methods. For token handling strategies (IGD, CFT, and Pos), we implement each on top of both BIGRec and D3 for comparison. For traditional baselines such as GRU4Rec and SASRec, we adopt the reported results from the D3 paper [19] to ensure consistency. For LRURec, we utilize the official PyTorch implementation [32] and evaluate it using our dataset split. Our key observations are as follows:

- IGD achieves notable improvements on both LLM4Rec methods (BIGRec and D3). Specifically, it yields an average improvement of **20.9%** in HR@10 and **21.5%** in NDCG@10 on BIGRec, and **18.9%** in HR@10 and **20.1%** in NDCG@10 on D3.

- Compared with other token-handling methods (Pos and CFT), IGD consistently delivers better performance across all evaluation metrics on both the LLM4Rec backbone, showing its effectiveness and generalizability. The superiority of the proposed IGD method can be attributed to its specific consideration of token decisiveness.

- LLM4Rec methods (BIGRec, D3) clearly outperform traditional recommendation models (GRU4Rec, SASRec). Although LRURec outperforms them on the CDs and Books datasets, BIGRec and D3 regain the lead when combined with token reweighting techniques, highlighting the advantages of leveraging LLMs in recommendation tasks.

Table 3: Ablation results of IGD on D3. Here, "w/o" $w_d$, "w/o $w_t$", and "w/o Both" denote the removal of decoding reweighting, tuning reweighting, and both components, respectively.

| Methods | CDs | | | | Games | | | |
|---|---|---|---|---|---|---|---|---|
| | N@5 | N@10 | H@5 | H@10 | N@5 | N@10 | H@5 | H@10 |
| IGD | 0.0748 | 0.0801 | 0.0929 | 0.1092 | 0.0518 | 0.0598 | 0.0705 | 0.0946 |
| w/o $w_d$ | 0.0751 | 0.0800 | 0.0926 | 0.1077 | 0.0514 | 0.0594 | 0.0695 | 0.0942 |
| w/o $w_t$ | 0.0718 | 0.0768 | 0.0887 | 0.1041 | 0.0414 | 0.0484 | 0.0575 | 0.0790 |
| w/o Both | 0.0716 | 0.0767 | 0.0882 | 0.1040 | 0.0415 | 0.0477 | 0.0581 | 0.0773 |

| Methods | Toys | | | | Books | | | |
|---|---|---|---|---|---|---|---|---|
| | N@5 | N@10 | H@5 | H@10 | N@5 | N@10 | H@5 | H@10 |
| IGD | 0.0658 | 0.0726 | 0.0868 | 0.1082 | 0.0291 | 0.0313 | 0.0356 | 0.0424 |
| w/o $w_d$ | 0.0653 | 0.0719 | 0.0861 | 0.1063 | 0.0290 | 0.0312 | 0.0355 | 0.0422 |
| w/o $w_t$ | 0.0640 | 0.0711 | 0.0843 | 0.1060 | 0.0212 | 0.0229 | 0.0268 | 0.0318 |
| w/o Both | 0.0634 | 0.0698 | 0.0833 | 0.1029 | 0.0212 | 0.0228 | 0.0266 | 0.0315 |

## 5.3 Ablation Study (RQ2)

In this section, we analyze the individual contributions of the two stages of IGD reweighting—Tuning and Decoding—to the overall improvement in recommendation accuracy. The results of the ablation study are presented in Table 3. We observe the following: (1) Both the tuning-stage and the decoding-stage reweighting contribute positively to performance gains; (2) The tuning stage brings a more substantial improvement than the decoding stage. Notably, even when using only the tuning-stage reweighting (w/o $w_d$), IGD still outperforms the CFT and Pos baselines.

## 5.4 In-depth Analysis of IGD's Effect (RQ3)

In this subsection, we investigate how IGD influences the underlying mechanisms of tuning and decoding, and how these effects contribute to the improvement of recommendation performance. We follow a similar experimental strategy as described in Section 3.3 to compare tuning loss and prefix entropy.

1. **IGD-Tuning**: As shown in Fig. 5, after incorporating the weight term $w_t$, the training loss of zero-IG tokens decreases more slowly, whereas the loss for non-zero-IG tokens decreases more rapidly. This indicates that IGD-Tuning encourages learning on non-zero-IG tokens while avoiding overfitting zero-IG ones.

2. **IGD-Decoding**: As illustrated in Figure 6, the entropy gap between the predicted and ground truth prefixes is reduced under IGD-Decoding, indicating a more decisive and better-aligned decoding process. Note that increasing $\alpha$ (see Equation (9)) to mitigate decoding bias does not always improve recommendation accuracy, suggesting a trade-off between maximizing the likelihood of the token sequence and prioritizing high-decisiveness tokens.

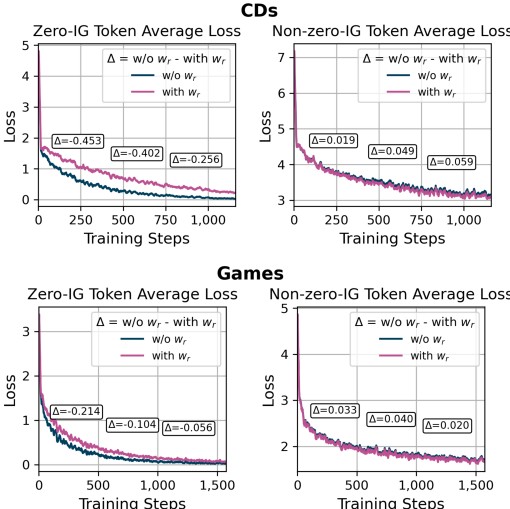

Figure 5: Loss comparison on CDs and Games datasets: IGD-Tuning effect on zero-IG and non-zero-IG tokens (epoch 1). The results on the other two datasets are in Appendix C.

## 5.5 Generalizability (RQ4)

To demonstrate the generalizability of our proposed IGD method, we evaluate it under different tokenization strategies and model scales. Specifically, we conduct experiments using the LLaMA3-8B model [36], with the prefix tree constructed using the LLaMA3 tokenizer [36]. To enable efficient

Table 4: Performance comparison of D3+IGD vs. original D3 on LLaMA3-8B backbone on CDs and Games. The results on the other two datasets are in Appendix E.

| Methods | CDs | | | | Games | | | |
|---------|-----|-----|-----|------|-------|-----|-----|------|
| | N@5 | N@10 | H@5 | H@10 | N@5 | N@10 | H@5 | H@10 |
| D3 | 0.0742 | 0.0790 | 0.0917 | 0.1062 | 0.0456 | 0.0528 | 0.0611 | 0.0832 |
| **+IGD** | **0.0791** | **0.0850** | **0.0994** | **0.1179** | **0.0645** | **0.0734** | **0.0863** | **0.1137** |
| *Improvement* | *+6.60%* | *+7.59%* | *+8.40%* | *+11.0%* | *41.5%* | *+39.0%* | *+41.2%* | *+36.7%* |

fine-tuning, we employ 4-bit QLoRA [37] for both training and inference, with the rank parameter $r$ set to 32, the scaling factor $\alpha$ set to 64, and the dropout rate set to 0.05. As shown in Table 4, IGD yields consistent improvements across all four datasets when applied to D3, with an average gain of 26.8% in HR@10 and 26.4% in NDCG@10. In addition to the Amazon dataset, we also evaluate IGD on the Steam dataset (see Appendix E), where it demonstrates consistent improvements. These results confirm the generalizability of our IGD approach across different model architectures and datasets.

# 6 Related Work

**LLM-based Recommendation:** Based on how LLMs are utilized, existing LLM4Rec methods can be broadly categorized into two groups: (1) using LLMs to augment traditional recommendation models [38, 39, 40], and (2) employing LLMs directly as recommendation systems [20, 19, 41]. Our work aligns with the second category. Early approaches in this direction typically followed a discriminative paradigm [24], which has since shifted toward a generative paradigm [20]. Building on this, recent studies have explored enhancing collaborative modeling [42, 43, 44], optimizing tokenization schemes [22], and improving inference efficiency [45], *etc*. However, few works have investigated recommendations at the token level. Our work specifically focuses on this.

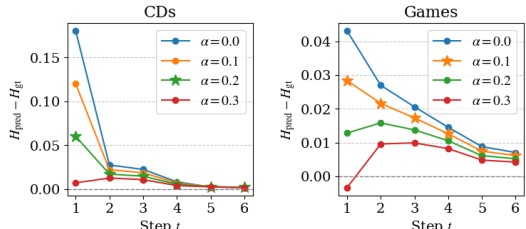

Figure 6: Entropy difference: prediction vs. ground truth after IGD-decoding. "Start" indicates optimal $\alpha$ selected based on HR@10. The results on the other two datasets are in Appendix D.

**Token-level Biases in LLM4Rec:** Autoregressive LLMs generate items token-by-token, misaligning with item-level recommendation objectives [46, 19, 47, 48]. Multi-token items suffer from token-level biases: high-probability *ghost tokens* dominate decoding without aiding discrimination, inflating scores for longer items [19]; and *common tokens* share across many items (e.g., 'The') dilute the impact of rare and informative tokens, reducing diversity [46, 19].

**Token-level Bias Mitigation Strategies:** To reduce amplification bias caused by *ghost tokens*, D3 removes length normalization in decoding [19], but *ghost tokens* still skew beam search. Position normalization reweights tokens by position, assuming early tokens are more uncertain [33], yet *ghost tokens* appear throughout sequences and are not confined to later positions. To prevent *common token* domination, some methods leverage traditional model guidance [47, 19], but this limits flexibility and scalability. Different from them, we propose a token-handling strategy by considering token decisiveness, which helps debiasing.

**Token Prefix Trie for LLM4Rec:** Recent works leverage token prefix trie for LLM-based recommendation, each employing the trie in a different way to guide token generation and learning. MSL [48] uses the trie as a constraint: a masked softmax prunes infeasible next tokens, reducing negative optimization signals and focusing training on valid continuations. Flower [47] uses the trie for process-guided supervision: rewards are stored at nodes and propagated along paths during tuning, encouraging trajectories that obtain higher rewards. Ours uses the trie for decisiveness-aware weighting: each node stores an IG score estimated from data, and these scores are applied to debias both tuning and decoding, highlighting discriminative tokens while down-weighting ambiguous ones.

# 7   Conclusion & Limitation

In this work, we introduced a decision-process perspective for token-by-token generation in LLM4Rec, quantifying token decisiveness using IG. We identified tuning and decoding biases where current models misallocate focus between decisive and non-decisive tokens. Our proposed IGD method addressed these issues through token reweighting during both training and inference. Experiments across four datasets, two LLM4Rec backbones, and two LLM architectures demonstrated IGD's effectiveness, achieving significant improvements in recommendation accuracy.

Our current experiments focus exclusively on IG-based token scoring. However, the IG values of tokens are highly skewed (see Figure 4), which makes it challenging to design practical reweighting methods (as discussed in Appendix G). Future work may explore alternative decision metrics, such as Gini impurity, Gain Ratio, or Chi-squared statistics [49, 50, 51], which may provide complementary perspectives on token decisiveness.

In addition, our present analysis is restricted to text tokens. Extending decisiveness modeling to semantic ID tokens [34, 52, 53] and multimodal tokens [54, 55, 56] is a promising direction for future research.

# 8   Acknowledgement

This research/project is supported by the National Research Foundation, Singapore under its National Large Language Models Funding Initiative (AISG Award No: AISG-NMLP-2024-002) and A*STAR under its Japan-Singapore Joint Call: Japan Science and Technology Agency (JST) and Agency for Science, Technology and Research (A*STAR) 2024 (Award R24I6IR142). Any opinions, findings and conclusions or recommendations expressed in this material are those of the author(s) and do not reflect the views of National Research Foundation, Singapore. Any opinions, findings and conclusions or recommendations expressed in this material are those of the author(s) and do not reflect the views of the A*STAR.

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

# A The Details of Compared Methods

To highlight the modeling strength of LLM4Rec models and provide context for our token-level enhancement method IGD, we first compare them with traditional sequential recommenders.

**Traditional Recommendation Methods:**

- **GRU4Rec** [30]: A widely-used sequential recommendation method that employs Gated Recurrent Units (GRU) to capture sequential patterns and model user preferences.

- **SASRec** [31]: A representative sequential recommendation method that utilizes a self-attention mechanism for preference modeling, offering powerful representation capabilities for sequential data.

- **LRURec** [32]: A linear recurrent unit-based sequential recommender that enables fast, incremental inference with reduced model size and parallelizable training, achieving strong accuracy and efficiency compared to attention-based baselines.

**LLM4Rec Methods:**

- **BIGRec** [20]: A representative LLM4Rec system that fine-tunes large language models to generate the next item based on the user's historical behavior. We adopt the constrained beam search decoding paradigm as described in D3 paper [19].

- **D3** [19]: A state-of-the-art LLM4Rec approach that fine-tunes the model similarly to BIGRec but mitigates amplification bias by removing length normalization during beam search decoding. In addition, it incorporates an ensemble design with traditional models, which we omit for a fair comparison.

**Compared Token Reweighting Methods**    To evaluate the effectiveness of IGD as a token reweighting strategy, we compare it against other token-level methods that can be seamlessly integrated into LLM4Rec frameworks. These methods include:

- **Position Normalization (Pos)** [33]: This method reweights tokens during SFT based on their position in the sequence, assigning higher weight to earlier tokens compared to later tokens.

- **Causal Fine-tuning (CFT)** [33]: Building upon the Pos method, CFT introduces an additional context-aware loss term. This loss captures the difference between contextual and non-contextual token predictions (representing causal effects [57]), encouraging the model to emphasize tokens that are more tightly correlated with the input context.

# B Implementation Details of Compared Method

For traditional baselines, we directly adopt the settings from the D3 paper [19], as our experimental setup is fully aligned with theirs. For LLM-based methods, we use Qwen2.5-1.5B [28] as the backbone. The batch size is set to 64, the optimizer is AdamW, the learning rate is $1 \times 10^{-4}$, and the dropout rate is 0.05. Model selection is based on validation loss, with an early stopping strategy that uses a maximum of 3 epochs and a patience of 1 epoch. All other settings follow the D3 paper. For our proposed IGD method, the tuning hyperparameter $\beta$ is selected from the set $\{0.08, 0.1, 0.2, 0.3, 0.4, 0.5, 0.6, 1.0\}$, and the decoding hyperparameter $\alpha$ is selected from $\{0.0, 0.1, 0.2, 0.3, 0.4\}$. We first search for the optimal $\beta$ based on validation performance, followed by a search for the best $\alpha$.

# C Loss on All Datasets

This section presents a comparison of the training loss for zero-IG and non-zero-IG tokens before and after applying our IGD-tuning across all datasets. The results are summarized in Figure 7.

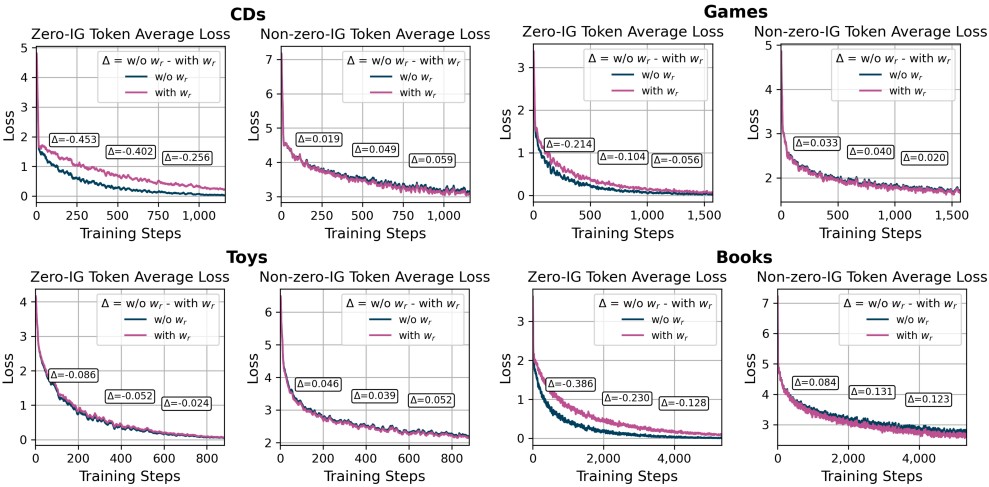

Figure 7: Loss comparison: IGD-Tuning effect on zero-IG and non-zero-IG tokens (epoch 1).

# D   Entropy difference on All Datasets

This section presents a comparison of entropy differentials across all datasets following the application of IGD-decoding. As the value of $\alpha$ increases, the decoding bias decreases. The results summarized in Figure 8

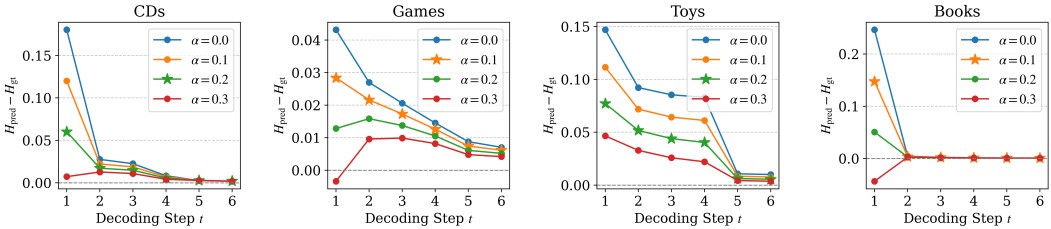

Figure 8: Entropy difference: prediction vs. ground truth after IGD-decoding. "Start" indicates optimal $\alpha$ selected based on HR@10 on all datasets.

# E   All Results for Generalizability Study

This section presents all the results for the Generalizability study. The results are summarized in Table 5 and Table 6

**More dataset.**   To further assess generalizability, we additionally evaluated the best-performing baseline (D3) and its IGD-enhanced variant on Steam dataset ($\sim$982K interactions). Results in Table 6 show that IGD continues to provide consistent gains over the baseline.

# F   Item Diversity under IGD-Decoding

We evaluate IGD-D's effect on item diversity with $\alpha \in \{0.0, 0.1, 0.2, 0.3, 0.4\}$.

**First Word Repetition Rate (FWR)**—lower is better—is the proportion of the most frequent first token among the top-10 recommended items. **Item Score Entropy (ISE)**—higher is better—is computed from the decoding probabilities induced by final item scores using the standard $\sum -p \log p$ aggregation.

As Table 7 is shown, increasing $\alpha$ consistently decreases FWR and increases ISE across all datasets.

Table 5: Performance comparison of D3+IGD vs. original D3 on LLaMA3-8B backbone across different recommendation datasets

| Methods | CDs | | | | Games | | | |
|---|---|---|---|---|---|---|---|---|
| | N@5 | N@10 | H@5 | H@10 | N@5 | N@10 | H@5 | H@10 |
| D3 | 0.0742 | 0.0790 | 0.0917 | 0.1062 | 0.0456 | 0.0528 | 0.0611 | 0.0832 |
| **+IGD** | **0.0791** | **0.0850** | **0.0994** | **0.1179** | **0.0645** | **0.0734** | **0.0863** | **0.1137** |
| *Improvement* | *+6.60%* | *+7.59%* | *+8.40%* | *+11.0%* | *41.5%* | *+39.0%* | *+41.2%* | *+36.7%* |

| Methods | Toys | | | | Books | | | |
|---|---|---|---|---|---|---|---|---|
| | N@5 | N@10 | H@5 | H@10 | N@5 | N@10 | H@5 | H@10 |
| D3 | 0.0739 | 0.0797 | 0.0916 | 0.1096 | 0.0198 | 0.0214 | 0.0249 | 0.0299 |
| **+IGD** | **0.0885** | **0.0932** | **0.1102** | **0.1311** | **0.0282** | **0.0304** | **0.0352** | **0.0418** |
| *Improvement* | *+19.8%* | *+16.9%* | *+20.3%* | *+19.6%* | *+42.4%* | *+42.1%* | *+41.4%* | *+39.8%* |

Table 6: Results on the Steam dataset ($\sim$982K interactions) under the same evaluation protocol as the main study.

| Method | HR@10 | NDCG@10 |
|---|---|---|
| D3 | 0.1126 | 0.0782 |
| +IGD | 0.1184 (+5.15%) | 0.0810 (+3.58%) |

Mechanism Analysis: Non-decisive tokens tend to receive high logits and dominate beam expansion, reducing diversity. IGD-D increases the logits of decisive tokens (via $\alpha$), yielding lower FWR and higher ISE.

## G  Why IGD-Tuning Adopts a Binary Weighting Scheme

Our binary-based weighting scheme is motivated by the observed behavior of IG in our setting and is not arbitrary. Two empirical observations guide the design:

**(1) Distinct zero-IG token group.** Tokens with zero IG form a dominant cluster (about 55% of all tokens) and exhibit disproportionately high logits with very low training loss. Treating this group as a separate class and applying a uniform down-weighting factor $\beta$ effectively mitigates their outsized influence.

**(2) Non-linear IG distribution among non-zero tokens.** The IG distribution for non-zero tokens is highly skewed and roughly exponential rather than linear. This makes it difficult to craft a smooth, well-calibrated continuous weighting function over IG. We experimented with a simple linear and monotonic alternative:

$$w_t \ = \ \beta \ + \ (1 - \beta) \cdot \frac{\text{IG}}{\text{IG}_{\max}}$$

As shown in Table 8, this linear function does not outperform the binary scheme.

Due to the dominance of zero-IG tokens and the non-linear nature of non-zero IG values, a binary separation with a calibrated $\beta$ provides stronger and more stable improvements than a linear mapping, across all evaluated datasets.

## H  Effective Hyperparameter Ranges

Our method introduces two parameters: a decoding weight $\alpha$ and a training-time weight $\beta$. Since $\alpha$ is easy to tune, we only analyze $\beta$'s sensitivity while fixing $\alpha$=0.0. Unless otherwise noted, the search grid for $\beta$ is $\{0.08, 0.1, 0.2, 0.3, 0.4, 0.5, 0.6\}$; after selecting $\beta$, $\alpha$ is tuned over $\{0.1, 0.2\}$.

**Optimal $\beta$.** CDs: $\beta$=0.2 (HR@10=0.1077). Games: $\beta$=0.2 (HR@10=0.0942). Toys: $\beta$=0.5 (HR@10=0.1063). Books: $\beta$=0.1 (HR@10=0.0422).

Table 7: Diversity vs. $\alpha$ across datasets. FWR↓ and ISE↑.

| | CDs | | Toys | | Games | | Books | |
|---|---|---|---|---|---|---|---|---|
| $\alpha$ | FWR↓ | ISE↑ | FWR↓ | ISE↑ | FWR↓ | ISE↑ | FWR↓ | ISE↑ |
| 0.0 | 0.364 | 2.65 | 0.588 | 2.67 | 0.340 | 2.95 | 0.382 | 2.75 |
| 0.1 | 0.332 | 2.70 | 0.577 | 2.73 | 0.328 | 2.98 | 0.339 | 2.81 |
| 0.2 | 0.304 | 2.76 | 0.564 | 2.80 | 0.321 | 3.01 | 0.306 | 2.87 |
| 0.3 | 0.282 | 2.84 | 0.557 | 2.88 | 0.307 | 3.04 | 0.278 | 2.93 |
| 0.4 | 0.265 | 2.92 | 0.551 | 2.95 | 0.296 | 3.08 | 0.259 | 2.99 |

Table 8: Comparison of linear vs. binary IGD-T weighting across datasets. Best $\beta$ (Linear): 0.4, 0.9, 0.8, 0.4 for CDs, Games, Toys, Books. Best $\beta$ (Binary): 0.2, 0.2, 0.5, 0.1 for CDs, Games, Toys, Books.

| Dataset | Metric | Baseline (D3) | IGD-T (Linear) | IGD-T (Binary, ours) |
|---|---|---|---|---|
| CDs | HR@10 | 0.1040 | 0.1072 | 0.1077 |
| | NDCG@10 | 0.0767 | 0.0793 | 0.0800 |
| Games | HR@10 | 0.0773 | 0.0800 | 0.0942 |
| | NDCG@10 | 0.0477 | 0.0492 | 0.0594 |
| Toys | HR@10 | 0.1029 | 0.1034 | 0.1063 |
| | NDCG@10 | 0.0698 | 0.0692 | 0.0719 |
| Books | HR@10 | 0.0315 | 0.0339 | 0.0422 |
| | NDCG@10 | 0.0228 | 0.0245 | 0.0312 |

**Effective ranges (HR@10 and NDCG@10 > baseline).** CDs: $[0.1, 0.6]$; Games: $[0.1, 0.4] \cup \{0.6\}$; Toys: $[0.2, 0.6]$; Books: $[0.1, 0.6]$.

**Observation.** Each dataset exhibits a broad interval where performance exceeds the baseline, making $\beta$ easy to integrate into existing methods. However, to obtain the optimal $\beta$, one still needs to search over a reasonable range.

