# OpenReview forum: "IGD: Token Decisiveness Modeling via Information Gain in LLMs for Personalized Recommendation"
_NeurIPS.cc/2025/Conference — NeurIPS 2025 poster_

### Official Review · Reviewer_YKBt · 2025-06-27

**Clarity:** 4
**Significance:** 3
**Originality:** 4
**Rating:** 5
**Confidence:** 5

**Summary:**

This paper proposes an Information Gain-based Decisiveness-aware Token Handling (IGD) strategy for Large Language Model-based Recommendation (LLM4Rec), targeting the issue that existing methods usually treat all item tokens equally, ignoring differences in their decisiveness. The authors treat the item generation process as a decision process and use Information Gain (IG) to quantify each token’s decisiveness. They observe that many tokens have low IG but high logits, introducing bias in both training and decoding. IGD addresses this by down-weighting low-IG tokens during tuning and decoding to emphasize high-IG tokens—shifting the focus from pure likelihood maximization to decision-critical tokens. Experiments on four benchmark datasets with two LLM backbones show that IGD consistently improves recommendation accuracy.

**Questions:**

Q1. Has the author considered a more fine-grained token categorization, beyond simply distinguishing between zero and non-zero information gain tokens? If so, what are the results?

Q2. What is the candidate item distribution? Is it determined by the frequency of item occurrences in the interaction dataset?

**Ethical Concerns:**

["NO or VERY MINOR ethics concerns only"]

**Final Justification:**

I have read the authors’ response and decided to maintain my positive assessment.

**Limitations:**

Yes

**Paper Formatting Concerns:**

I didn't find any major formatting issues in this paper.

**Quality:**

3

**Strengths And Weaknesses:**

Strengths:

S1. Measuring the importance of item tokens from the perspective of “decisiveness” is interesting and introduces a novel angle.

S2. The authors treat the item generation process as a decision-making process and define token decisiveness through information gain. This definition is both novel and well-justified.

S3. The proposed method is applicable during both the tuning and inference stages, allowing it to emphasize high-decisiveness tokens throughout the modeling pipeline.

S4. The paper is well-motivated: it begins with a preliminary experiment to illustrate the problem, followed by the introduction of the proposed method.

S5. Extensive experiments on four benchmark datasets, using two LLM backbones (Qwen2.5 and LLaMA3), demonstrate consistent performance improvements.

Weaknesses:

W1. Currently, tokens are divided into only two categories based on information gain. While this binary classification is reasonable, a more fine-grained categorization—guided by the magnitude of information gain—may lead to a better results.

W2. Figure 1 is difficult to follow. Additionally, the layout of Section 3 could be made more compact to improve clarity and readability.

W3. The candidate item distribution is not clearly explained and requires further explanation.

W4. A concurrent work [1] also models item generation as a decision tree, which bears some relevance to this paper. A discussion of this related work should be included to clarify distinctions and highlight contributions.

[1] Gao et al. Process-Supervised LLM Recommenders via Flow-guided Tuning.

---

> ### Author Rebuttal · Authors · 2025-07-31
>
> We sincerely thank the reviewer for their positive assessment and insightful feedback on our manuscript. We are encouraged that the reviewer found our work to be novel, well-motivated, and supported by extensive experiments (S1–S5). We appreciate the constructive suggestions, which have helped us improve the clarity and rigor of our paper. Below, we address each of the reviewer's concerns and questions in detail.
>
> ---
>
> ### W1&Q1: Why not take a more fine-grained weighting?
> A1: Thank you for the question. Our binary-based weighting design is motivated by two key observations about the behavior of Information Gain (IG) in our preliminary experiments:
>
> 1) Distinct Zero-IG Token Group: As shown in Figures 2 and 4, tokens with zero IG form a distinct and dominant group in the dataset (>55% of all tokens). These tokens consistently exhibit extremely low training loss and disproportionately high logits. Treating them as a separate class and applying a uniform down-weighting factor $\beta$ is a natural and effective way to handle their outsized influence.
>
> 2) Non-linear IG Distribution: As illustrated in Figure 4, the distribution of non-zero IG values is highly skewed and roughly exponential, not linear. This makes it difficult to design a meaningful continuous function over IG values that results in well-calibrated weights. We experimented with a simple linear and monotonic weighting function:
> $$w_t = \beta + (1-\beta) \cdot \frac{\mathrm{IG}}{\mathrm{IG}_{\max}}$$
> However, as shown in the following tables, this linear function does not outperform our binary approach.
>
> | Dataset | Metric   | Baseline | IGD-T (Linear) | IGD-T (Binary, ours) |
> |---------|----------|----------|----------------|----------------------|
> | CDs     | HR@10    | 0.1040   | 0.1072         | 0.1077               |
> |         | NDCG@10  | 0.0767   | 0.0793         | 0.0800               |
> | Games   | HR@10    | 0.0773   | 0.0800         | 0.0942               |
> |         | NDCG@10  | 0.0477   | 0.0492         | 0.0594               |
> | Toys    | HR@10    | 0.1029   | 0.1034         | 0.1063               |
> |         | NDCG@10  | 0.0698   | 0.0692         | 0.0719               |
> | Books   | HR@10    | 0.0315   | 0.0339         | 0.0422               |
> |         | NDCG@10  | 0.0228   | 0.0245         | 0.0312               |
>
> *(IGD-T (Linear): best β=0.4, 0.9, 0.8, 0.4 for CDs, Games, Toys, Books)*
>
> *(IGD-T: best β=0.2, 0.2, 0.5, 0.1 for CDs, Games, Toys, Books)*
>
> ---
>
>  ### W2: Clarity of Figure 1 and Section 3
>  A2: Thanks for pointing this out. We will revise the figure to improve its clarity and update Section 3 accordingly.
>
> ---
>
> ### W3 & Q2: What is the candidate item distribution? Is it determined by the frequency of item occurrences in the interaction dataset?
> A3: Yes. The candidate item distribution, denoted as $P(\mathcal{I})$, represents the prior probability of any item  $\mathcal{I}$, estimated by the frequency of item occurrences in the interaction dataset.
>
> ---
>
> ### W4. Discussion of Concurrent Work
> A4: Thanks for bringing this highly relevant concurrent work to our attention. While both our work and Gao et al. conceptualize item generation as a sequential decision-making process, our approaches differ fundamentally in their problem formulation, methodology, and goals:
>
> - **Core Idea and Methodology:**
>   - **Flow-guided Tuning (Gao et al.):** They introduce a "Flow-guided process reward" to supervise the entire sequence of token generation logits, rewarding paths that are more efficient at identifying the target item. Their contribution is a novel process-level supervision signal.
>   - **Our Work (IGD):** We start from an information-theoretic perspective to identify a fundamental issue: a discrepancy between a token's generation likelihood and its actual "decisiveness" in narrowing down the candidate set. We propose Information Gain (IG) to quantify this decisiveness and identify two specific biases ("Tuning Bias" and "Decoding Bias"). Our IGD strategy is a targeted intervention that re-weights token losses during tuning and adjusts logits during decoding to mitigate these biases.
>
> We hope this discussion clarifies the unique contributions of our work and its relationship to concurrent research. In the revised manuscript, we have added the discussion to the "Related Work".

---

> > ### Comment · Reviewer_YKBt · 2025-08-03
> >
> > Thanks for the response. I will maintain my positive rating.

---

### Official Review · Reviewer_14G4 · 2025-07-03

**Clarity:** 3
**Significance:** 2
**Originality:** 3
**Rating:** 4
**Confidence:** 3

**Summary:**

This work improves LLM-based recommendation systems, which traditionally treat all tokens equally and maximize likelihood, by introducing a reweighting approach that prioritizes more informative tokens based on their importance. Experiments were conducted on four datasets and two backbone models, demonstrating substantial improvements in recommendation performance.

**Questions:**

1. Compared to previous models, this approach seems to involve more computation. Could you share approximately how much training and decoding time it requires in practice?
2. During decoding, as the IG weighting increases, it seems likely that the model would increasingly prioritize informative tokens. Did you observe any issues with reduced diversity or over-concentration on certain items as a result?

**Ethical Concerns:**

["NO or VERY MINOR ethics concerns only"]

**Final Justification:**

The rebuttal effectively addressed my concerns regarding diversity and provided additional experiments on a new dataset. The proposed token decisiveness-based training and decoding strategy is novel and highlights the potential need for similar decoding mechanisms in other LLM-based recommendation models. The reported computation overhead seems generally reasonable in offline batch settings, though its real-time latency impact remains less explored; this does not strongly affect my current recommendation. Given these clarifications and improvements, I recommend borderline accept.

**Limitations:**

Yes, the authors have discussed the main limitations of their work. However, I think the scalability to real-time applications could also be acknowledged.

**Paper Formatting Concerns:**

Not a major issue, but I noticed that page 4 contains quite a lot of empty space. It could be helpful to fill that area with additional explanation or commentary for the equations.

**Quality:**

3

**Strengths And Weaknesses:**

- **Quality**: The paper clearly identifies a key limitation of existing LLM-based recommendation systems and proposes an appropriate solution. However, the approach appears computationally intensive and likely to incur significant training and inference time.
- **Clarity**: The research idea, mathematical formulations, and experimental results are all described clearly and precisely.
- **Significance**: The paper proposes a practical improvement to the bias issue in token-level generation and convincingly demonstrates strong performance through extensive experiments. However, since the evaluation is limited to Amazon datasets, it would be important to validate the approach in other domains.
- **Originality**: The token decisiveness-based training and decoding strategy is novel. It highlights the potential need for similar decoding mechanisms in existing LLM-based recommendation models.

---

> ### Author Rebuttal · Authors · 2025-07-31
>
> We sincerely thank the reviewer for their thoughtful and constructive feedback. We are grateful for the positive assessment of our work's quality, clarity, and originality. The reviewer's insightful questions have prompted us to conduct further analyses that we believe have significantly strengthened our paper.
>
> Below, we address each of the reviewer's points with detailed, quantitative evidence.
>
>
> ---
>
> ### W1: This approach seems to involve more computation. How much training and decoding time it requires in practice?
> A1: During training, the additional cost of our method comes primarily from looking up the IG score for each token (performed via a prefix trie) and applying multiplicative weighting. This overhead is relatively small. During decoding, the extra cost involves looking up IG scores and reweighting the logits of candidate tokens based on these scores. This process is also lightweight.
> We summarize the empirical comparison results in the table below. As shown, the additional cost introduced by our method is generally acceptable.
> | Stage    | Overhead |
> |----------|----------|
> | Tuning   | +3.35%   |
> | Decoding | +9.79%   |
> (Test using Toys dataset with H100GPU)
>
>
> ---
>
> ### W2: Evaluation on datasets beyond Amazon domain.
> A2: Thank you for the suggestion. We have conducted an additional experiment comparing the best-performing baseline (D3) with our IGD method (applied to D3) on a new dataset from Steam, which contains approximately 982K interactions. The results are summarized in the following table. As shown, our method continues to yield meaningful performance improvements.
> Notably, due to time constraints, we have not included all baselines in this comparison. We will include the full set of results in the revised version of the paper.
>
>
> | Method    | HR@10           | NDCG@10         |
> |-----------|-----------------|-----------------|
> | Baseline  | 0.1126          | 0.0782          |
> | IGD       | 0.1184 (+5.15%) | 0.0810 (+3.58%) |
>
> ---
>
> ### W3: On Decoding Diversity: As the IG weighting increases, it seems likely that the model would increasingly prioritize informative tokens. Is there any issue with reduced diversity or over-concentration on certain items?
>
> A3: Thank you for this insightful question. Contrary to the potential concern, our proposed method, IGD-D, does not reduce diversity. In fact, our empirical results consistently show that **as the weight on informative tokens (controlled by `α`) increases (in a wide range), the diversity of the recommended items also increases.**
>
> **Empirical Verification**:
>
> We evaluated this effect using two diversity metrics across four standard datasets:
> - **item_score_entropy** (higher is better; indicates more diverse recommendations)
> - **first_word_repetition_ratio** (lower is better; indicates less repetition at the start of recommended items)
>
> The results below demonstrate a clear trend across all datasets. As `α` increases from 0.0 to 0.4: `item_score_entropy` consistently rises and `first_word_repetition_ratio` consistently falls:
>
> **CDs**
>
> | α     | item_score_entropy | first_word_repetition_ratio |
> |-------|--------------------|----------------------------|
> | 0.0 | 2.65               | 0.364                      |
> | 0.1 | 2.70               | 0.332                      |
> | 0.2 | 2.76               | 0.304                      |
> | 0.3 | 2.84               | 0.282                      |
> | 0.4 | 2.92               | 0.265                      |
>
> **Toys**
>
> | α     | item_score_entropy | first_word_repetition_ratio |
> |-------|--------------------|----------------------------|
> | 0.0 | 2.67               | 0.588                      |
> | 0.1 | 2.73               | 0.577                      |
> | 0.2 | 2.80               | 0.564                      |
> | 0.3 | 2.88               | 0.557                      |
> | 0.4 | 2.95               | 0.551                      |
>
> **Video Games**
> | α     | item_score_entropy | first_word_repetition_ratio |
> |-------|--------------------|----------------------------|
> | 0.0 | 2.95               | 0.340                      |
> | 0.1 | 2.98               | 0.328                      |
> | 0.2 | 3.01               | 0.321                      |
> | 0.3 | 3.04               | 0.307                      |
> | 0.4 | 3.08               | 0.296                      |
>
> **Books**
> | α     | item_score_entropy | first_word_repetition_ratio |
> |-------|--------------------|----------------------------|
> | 0.0 | 2.75               | 0.382                      |
> | 0.1 | 2.81               | 0.339                      |
> | 0.2 | 2.87               | 0.306                      |
> | 0.3 | 2.93               | 0.278                      |
> | 0.4 | 2.99               | 0.259                      |
>
> **Analysis of the Mechanism:**
> - Root Cause: From our analysis (Figures 3 and 4), the model overly prioritizes non-informative tokens (such as "The" at the start of many item titles). These tokens tend to receive very high logits, leading beam search to preferentially expand paths containing them. As a result, the generated item sequences often share similar prefixes, which in turn reduces diversity and causes significant repetition.
>
> - Role and Effect of `IGD-D`:  Our `IGD-D` decoding strategy directly combats this issue by boosting the logits of informative tokens (i.e., setting `α > 0`). This adjustment encourages the model to prioritize more informative tokens during decoding, thus mitigating the inherent bias toward repetitive, non-informative outputs. Figure 6 ("Entropy difference: prediction vs. ground truth after IGD-decoding") has evaluated the effect from the item entropy perspective.

---

> > ### Comment · Reviewer_14G4 · 2025-08-08
> >
> > Thank you for the detailed answers to my questions and concerns. The reported overhead seems generally reasonable in the tested offline batch setting, though its impact in latency-sensitive real-time environments could be further examined. The concern regarding diversity has been well addressed, and I appreciate the additional experiments on a new dataset. I will maintain my positive score accordingly.

---

> ### Author Response · Authors · 2025-08-08
> **Request to Review Our Rebuttal Before Discussion Ends**
>
> Dear Reviewer,
>
> With less than 24 hours remaining for the author–reviewer discussion, we would greatly appreciate it if you could take a few minutes to read our rebuttal and consider adjusting your rating.
>
> Thank you, Authors

---

> ### Author Response · Authors · 2025-08-08
> **Thank You for Your Reply**
>
> Dear Reviewer,
>
> Thank you for the follow-up. We are glad our response addressed your concerns. We appreciate your time and support for our work.
>
> Best regards,
>
> The Authors

---

### Official Review · Reviewer_QLsy · 2025-07-05

**Clarity:** 2
**Significance:** 2
**Originality:** 3
**Rating:** 4
**Confidence:** 4

**Summary:**

The paper introduces a novel method on using large language models (LLMs) for recommendation. In particular, the authors study the entropy / information gain (IG) of different type of tokens and found that existing methods can cause the model to over-focus on zero-IG tokens while under-emphasizing non-zero-IG tokens during tuning. To address this, the authors model the item-generation process as a sequential decision task and weight the loss of each token in training depending on its IG. In addition, the authors also modify the generation process by revising the scores based on their IG, and therefore mitigating the bias toward high-logit but low-decisiveness tokens. The authors experiment on different datasets and compare to multiple benchmarks, where the proposed method can be applied to different eixsting approaches and show performance improvements.

**Questions:**

1. How does the proposed method apply to different decoding paradigms like sampling / constrained beam search, or does it work with a more genrealized sampling setting with different logits processing (top-k, top-p, repetition penalty etc.)?

2. Is the proposed method sensitive to differen decoding settings? Especially with key parameters like temperature.

**Ethical Concerns:**

["NO or VERY MINOR ethics concerns only"]

**Final Justification:**

Thanks for the response provided by the authors, the additional results do address most of my concerns of the paper. I also believe this setting can be extended for language generation tasks with similar techniques (e.g., based on token entropy etc.), but the current setting and experiment results are satisfactory. Considering the rebuttal efforts and the paper's merits / limitations, I have raised my score to 4 accordingly.

**Quality:**

2

**Strengths And Weaknesses:**

Pros:

1. The proposed method is interesting and studies an important problem from the information theory perspective, i.e., how IG / entropy affect the decoding process and influcenes the downstream task performance.

2. The proposed method is somewhat intuitive and reasonable, i.e., non-zero IG tokens with low logit values can be more effective to optimize in trainning. In addition, this method is simple and can be applied to different existing methods such as BIGRec and D3.

3. The authors experimented on multiple datasets, demonstrating its performance over different baseline methods.

Cons:

1. The authors only experimented on the RecSys setting with limited datasets, although this setting should naturally extend to more generaly language modeling tasks.

2. The proposed method intriduced additional hyperparameter complexity in both training and decoding (e.g., beta and alpha), increasing the hyperparameter searching costs while only providing marginal gains in many cases.

3. The selected RecSys datasets are all from amazon and rather small in sizes. I encourage the authors to experiment LLMs on larger datasets, potentially with multimodal inputs to comprehensively evaluate the proposed method. More baseline methods from both traditional sequential methods (LRURec etc.) and LLM methods (TallRec etc.) shouls also improve the quality of the experiments.

---

> ### Author Rebuttal · Authors · 2025-07-31
>
> We thank the reviewer for their insightful feedback and constructive suggestions. We appreciate that they found our method 'interesting' and 'intuitive', and the research problem 'important. To address their concerns about the method's scope, we have conducted several new experiments regarding hyperparameter complexity, dataset scale, and baseline comparisons. We believe these additions substantially strengthen the manuscript.
>
> ---
>
> ### 1. Can we naturally extend experiments to more generally language modeling tasks?
> A1: We would like to clarify that our proposed IGD method is specifically tailored for LLM-based recommendation. Its computation of Information Gain (IG) depends on a predefined, finite set of items and their frequencies—an assumption that does not hold in general language modeling tasks. This limitation makes it challenging to define item-set entropy or derive meaningful IG values in open-domain settings. Consequently, our method is not directly applicable to general language modeling at this stage. Nevertheless, we are interested in exploring extensions to broader domains in future work.
>
> ---
>
> ### 2. Hyperparameter Complexity
> A2: Though our method introduces two additional parameters, $\alpha$ and $\beta$, the parameter $\alpha$ is used only during decoding and is tuned over {0.1, 0.2} after selecting a suitable $\beta$, avoiding grid search. Moreover, as shown in our ablation results, even without using $\alpha$, our method still yields performance improvements. Therefore, tuning $\alpha$ can just incur minimal hyperparameter tuning cost.
>
> The primary tuning cost comes from $\beta$, which influences the training process. While we acknowledge that tuning $\beta$ introduces some overhead, our sensitivity analysis indicates that this cost is modest in practice. Across various datasets, our method consistently outperforms baselines over a broad range of $\beta$ values (as shown in the following table). This suggests that identifying a suitable $\beta$ for achieving performance gains is relatively straightforward and does not require extensive tuning.
>
> | Dataset | Baseline HR@10 (β=1.0) | Optimal β | Max HR@10 (within effective β) | Min HR@10 (within effective β) | Effective β Range (HR@10 & NDCG@10 > Baseline) |
> |:--------|:----------------------:|:---------:|:--------------------------:|:--------------------------:|:----------------------------------------------:|
> | CDs     |         0.1040         |    0.2    |      0.1077 (β=0.2)        |      0.1066 (β=0.4)        |                  [0.1, 0.6]                   |
> | Games   |         0.0773         |    0.2    |      0.0942 (β=0.2)        |      0.0777 (β=0.6)        |                [0.1, 0.4] $\cup$ [0.6]                |
> | Toys    |         0.1029         |    0.5    |      0.1063 (β=0.5)        |      0.1038 (β=0.2)        |                  [0.2, 0.6]                    |
> | Books   |         0.0315         |    0.1    |      0.0422 (β=0.1)        |      0.0335 (β=0.6)        |                  [0.1, 0.6]                    |
>
> ---
>
> ### 3. More dataset (dataset beyond Amazon) and more baselines (LRURec, TALLRec)
> A3: Thanks you for the suggestion. We response the two points, respectively.
>
> **More dataset:** Following your advice, we conducted an additional experiment comparing the best-performing baseline (D3) with our IGD method (applied to D3) on a new dataset from Steam, which contains approximately 982K interactions. The results are summarized in the following table. As shown, our method continues to deliver meaningful performance improvements.
>
> | Method    | HR@10           | NDCG@10         |
> |-----------|-----------------|-----------------|
> | Baseline  | 0.1126          | 0.0782          |
> | IGD       | 0.1184 (+5.15%) | 0.0810 (+3.58%) |
>
> **More baselins:** We have added two strong baselines: LRURec, the suggested traditional sequential recommender method, and Tiger, a recent generative recommendation method. We did not include TALLRec in the comparison, as it is not designed for the all-ranking setting used in our experiments. Due to time constraints, we conduct the comparison on the CDs dataset only. The results, presented in the following table, show that our method (applied to the D3 baseline) consistently outperforms these strong baselines.
>
> | Method   | HR@5   | NDCG@5 | HR@10  | NDCG@10 |
> |----------|--------|--------|--------|---------|
> | LRURec   | 0.0680 | 0.0540 | 0.0824 | 0.0586  |
> | Tiger    | 0.0748 | 0.0620 | 0.0868 | 0.0659  |
> | IGD      | 0.0929 | 0.0748 | 0.1092 | 0.0801  |
>
> *(We will include the full tables for all datasets in the revised manuscript.)*
>
> ---
>
> ### 4. Can the proposed method be applied to different decoding paradigms or generalized sampling settings with various logits processing strategies? Additionally, how sensitive is the method to decoding settings, such as temperature?
>
> A4:  First, we would like to clarify that our work focuses on recommendation tasks, where the model is required to generate a list of real-world items in each prediction. To achieve this, deterministic constrained beam search (do_sample=False) is commonly adopted, and we follow this setting as well. Under this setup, randomness introduced by sampling is eliminated. While in theory our method can be integrated with various decoding paradigms and sampling strategies, the specific requirements of recommendation systems make deterministic constrained beam search a natural and practical default choice in our framework.
>
> Regarding the sensitivity of our method to decoding settings, we conducted additional experiments using stochastic sampling (do_sample=True) and varied the temperature parameter. We report representative results under nearly deterministic (T = 0.1), low (T = 0.7), and default (T = 1.0) temperature settings by comparing our IGD versions with or without applying the proposed decoding strategy. The results are summarized in the table below.
>
> | Dataset | Method         | Deterministic | T=0.1  | T=0.7  | T=1.0  |
> |:--------|:---------------|:-------------:|:------:|:------:|:------:|
> | CDs     | IGD (w/o $w_d$)  |    0.1077     | 0.1076 | 0.1041 | 0.1015 |
> | CDs     | IGD (α=0.2)    |    0.1092     | 0.1085 | 0.1058 | 0.0951 |
> | Games   | IGD (w/o $w_d$)  |    0.0942     | 0.0934 | 0.0914 | 0.0914 |
> | Games   | IGD (α=0.1)    |    0.0946     | 0.0947 | 0.0926 | 0.0900 |
> | Toys    | IGD (w/o $w_d$)  |    0.1063     | 0.1052 | 0.1054 | 0.1018 |
> | Toys    | IGD (α=0.2)    |    0.1082     | 0.1078 | 0.1061 | 0.1032 |
> | Books   | IGD (w/o $w_d$)  |    0.0422     | 0.0423 | 0.0411 | 0.0390 |
> | Books   | IGD (α=0.1)    |    0.0424     | 0.0424 | 0.0410 | 0.0388 |
>
>
> As shown, our proposed decoding method $w_d$ remains robust and continues to bring performance Improvements (compared to our IGD without using the proposed decoding strategy)— at lower temperatures (T ≤ 0.7).

---

> ### Author Response · Authors · 2025-08-08
> **Request for Feedback on Our Rebuttal**
>
> Dear Reviewer,
>
> We noticed that your rating for our paper is currently hidden, yet we have not received any comments from you. Could you please share your feedback to let us know whether our rebuttal has addressed your concerns?
>
> Thank you,
> Authors

---

> > ### Comment · Reviewer_QLsy · 2025-08-08
> >
> > Dear authors, as mentioned in the final justification, the rebuttal & additional results address most of my concerns of the paper, I have raised my score overall score to 4 taking both the paper and the rebuttal into consideration.

---

> > > ### Author Response · Authors · 2025-08-09
> > > **Thank You for Your Reply**
> > >
> > > Dear Reviewer
> > >
> > > Thank you for your follow-up comment. We are glad to hear that our rebuttal and additional results have addressed your concerns. We appreciate your valuable feedback.
> > >
> > > Best regards,
> > >
> > > Authors

---

### Official Review · Reviewer_v8sa · 2025-07-07

**Clarity:** 3
**Significance:** 2
**Originality:** 3
**Rating:** 4
**Confidence:** 4

**Summary:**

The paper proposed a re-weighting method for both tuning and decoding for LLM4Rec setting (directly use LLM as recommender to generate next-item), based on the insight the some tokens doesn’t reduce the entropy of the to-be-generated item but are put on high focus in both tuning and decoding. Specifically, the paper proposed a metric called information gain to determine the importance of the token, and put larger weight on high importance token while lower weight on less important ones. The paper performs experiments on benchmark data and achieves good results.

**Questions:**

The method is simple and follows the intuition. My main concern is the setup and motivation.
From equation 5, during SFT, the model will focus more on the token that can dramatically decrease the entropy for token output, and the entropy is calculated using the train dataset.
From Table 1, the test dataset is very small compared to the train data, it might be hard to generalize the w_t if test data is large (i.e. in practice, testing data is much more compared to SFT data). In this case, the estimation of IG on training data can be very different from the actual potential distribution. Moreover, if during testing, there is a new item with a new title, then this method might not work. As in 5.1 first paragraph mentioned, the dataset is truncated in a way that each user and item has at least 5 interactions. Although this is used by another paper (one of the baseline), this kind of process actually leads to implicit leakage since this ensures the statistics in the training dataset can be leveraged to test the dataset.


In eq (7): instead of introducing a heuristic hyperparameter beta, why not make the weight a continuous and monotonic function of IG?
How to tune beta and alpha?


why are there tuning bias and decoding bias? more in-depth analysis on why this is the case would be helpful

**Ethical Concerns:**

["NO or VERY MINOR ethics concerns only"]

**Final Justification:**

I have read the rebuttal, while most concerns are addressed, I'm not convinced on the real performance since the evaluation dataset is unrealistic with 5-core filtering; however I acknowledge this is standard practice in the research community, and this paper does bring some novel ideas and insights that might inspire more follow-up work. Hence I'm willing to increase my score to 4.

**Limitations:**

The experiment setup does not reflect the real recommendation scenarios. See above comments.

the weighting approach seems highly heuristic but not mathematically motivated, and not clear how to tune the hyperparameters and no studies on the sensitivity of those

**Quality:**

3

**Strengths And Weaknesses:**

Strength:
* The paper has simple intuition and the experiment results are promising on several datasets.
* The proposed IG metric is simple to calculate. The overall algorithm is easy to be plugged into existing methods.

Weakness:
* The insight seems great, but the approach is quite heuristic but not mathematically motivated; it’s not clear how to tune the hyperparameters beta and alpha, and there is no empirical studies on the sensitivity of such hyperparameters
* The IG metric is specifically pre-defined through training dataset, and might be hard to generalize in real recommendation scenarios.
* The experiment are not convincing regarding to the IG metric, specifically the setup seems biased towards to proposed method (i.e. removing tail item to make sure training and testing distribution are more similar/similar type recommendation(i.e. CD, Games, etc)).

---

> ### Author Rebuttal · Authors · 2025-07-31
>
> We thank the reviewer for their constructive feedback. As the questions and weaknesses are closely related, we summarize the concerns and address them collectively, rather than responding to each point in the two sections separately.
>
> ---
>
> ### Q1. Unrealistic Experimental Setup
>
> Q1.1: The current data split, where the test set is much smaller than the training set, may limit the effectiveness of the evaluation.
>
> A1.1: We respectfully clarify that our data splitting strategy is designed to reflect common real-world recommendation scenarios.
> In many industry applications, models are trained on a large corpus of historical user interactions (the "training set") to predict user behavior over a shorter, subsequent time period (the "test set"). Our chronological 8:1:1 train-validation-test split directly follows this paradigm and is consistent with standard practice in existing works.
>
> However, to further address your concern, we conducted an additional experiment by  re-splitting the dataset into a 5:1:4 ratio for training, validation, and testing, with keeping the chronological order. This setup results in a test set comparable in size to the training set and introduces a significant temporal gap between training and testing periods. We compared our method, IGD-Tuning (IGD-T), with the strongest baseline (D3) under this new split. The results, summarized in the following table, show that IGD-T consistently and significantly outperforms D3. This demonstrates that our method (and the computed weights $w_t$) generalizes well and is not simply benefiting from a specific data split.
>
>
> | Dataset | Metric   | Baseline (D3) | IGD-T (ours) | Improvement |
> |---------|----------|---------------|--------------|-------------|
> | CDs     | HR@10    | 0.0923        | 0.0975       | +5.6%       |
> |         | NDCG@10  | 0.0698        | 0.0732       | +4.9%       |
> | Games   | HR@10    | 0.0680        | 0.0809       | +19.0%      |
> |         | NDCG@10  | 0.0452        | 0.0542       | +19.9%      |
> | Toys    | HR@10    | 0.1093        | 0.1141       | +4.4%       |
> |         | NDCG@10  | 0.0791        | 0.0822       | +3.9%       |
> | Books   | HR@10    | 0.0424        | 0.0588       | +38.7%      |
> |         | NDCG@10  | 0.0321        | 0.0447       | +39.3%      |
>
> ---
>
> Q1.2.  Data preprocessing introduces potential information leakage, as it allows the statistics in the training dataset can be leveraged to test the dataset
>
> A1.2: This appears to be a misunderstanding. Our data preprocessing does not suffer from information leakage. On the contrary, we adopt a chronological data splitting strategy specifically to avoid information leakage from the future into the past.
>
> The reviewer's concern seems primarily to stem from the use of 5-core filtering. First, we would like to clarify that this is not a filtering step introduced by any particular baseline, but rather a widely adopted standard practice in recommendation research. This approach is used in many well-known works in the field:
> - "Hidden factors and hidden topics" (McAuley & Leskovec, RecSys 2013): Established the use of 5-core filtering on the now-standard Amazon dataset.
> - "Image-based recommendations on styles and substitutes" (McAuley et al., SIGIR 2015): Continued the practice with 5-core filtering for visual recommendation.
> - "Neural Graph Collaborative Filtering" (Wang et al., SIGIR 2019): Adopted a 10-core filter for modern graph-based models, confirming k-core is a standard procedure.
>
> Second, we respectfully argue that applying 5-core filtering does not leak training data statistics into the test set. To illustrate this, consider the cold-start item scenario raised by the reviewer: the 5-core filtering is applied to the entire dataset before the chronological data split. As a result, items that appear for the first time during the test period are retained. Consequently, our test sets still include a substantial number of cold-start items—approximately 10% in the Books dataset and 8% in the Toys dataset—which naturally results in distributional drift from training to test. Additionally, significant drift can also be observed from the perspective of item frequency distribution.
>
> Table: Frequency Draft - Top 10% frequent item in the training set and their Avg. Freq. Percentile in Test set.
> | Category | Avg. Freq. Percentile in Test Set |
> |:---------|:---------------------------------------|
> | CDs_and_Vinyl | 31.52%                                 |
> | Video_Games | 30.67%                                 |
> | Toys_and_Games | 39.72%                                 |
> | Books    | 64.68%                                 |
>
> ---
>
> ### Q2: The Design of the IGD Token Weighting Scheme: Why not use a continuous and monotonic function of IG instead of a binary weighting approach?
>
> A2: Our binary-based weighting approach is not designed arbitrarily—it is motivated by two key observations about the behavior of Information Gain (IG) in our context:
>
> 1) Distinct Zero-IG Token Group: As shown in Figures 2 and 4, tokens with zero IG form a distinct and dominant group in the dataset (55% of all tokens). These tokens consistently exhibit extremely low training loss and disproportionately high logits. Treating them as a separate class and applying a uniform down-weighting factor $\beta$ is a natural and effective way to handle their outsized influence.
>
> 2) Non-linear IG Distribution: As illustrated in Figure 4, the distribution of non-zero IG values is highly skewed and roughly exponential, not linear. This makes it difficult to design a meaningful continuous function over IG values that results in well-calibrated weights. We experimented with a simple linear and monotonic weighting function:
> $$w_t = \beta + (1-\beta) \cdot \frac{\mathrm{IG}}{\mathrm{IG}_{\max}}$$
> However, as shown in the following tables, this linear function does not outperform our binary approach.
>
> | Dataset | Metric   | Baseline | IGD-T (Linear) | IGD-T (Binary, ours) |
> |---------|----------|----------|----------------|----------------------|
> | CDs     | HR@10    | 0.1040   | 0.1072         | 0.1077               |
> |         | NDCG@10  | 0.0767   | 0.0793         | 0.0800               |
> | Games   | HR@10    | 0.0773   | 0.0800         | 0.0942               |
> |         | NDCG@10  | 0.0477   | 0.0492         | 0.0594               |
> | Toys    | HR@10    | 0.1029   | 0.1034         | 0.1063               |
> |         | NDCG@10  | 0.0698   | 0.0692         | 0.0719               |
> | Books   | HR@10    | 0.0315   | 0.0339         | 0.0422               |
> |         | NDCG@10  | 0.0228   | 0.0245         | 0.0312               |
>
> *(IGD-T (Linear): best β=0.4, 0.9, 0.8, 0.4 for CDs, Games, Toys, Books)*
>
> *(IGD-T: best β=0.2, 0.2, 0.5, 0.1 for CDs, Games, Toys, Books)*
>
> ---
>
> ### Q3. Hyperparameter Sensitivity Analysis
> A3: Our method introduces two additional parameters, $\alpha$ and $\beta$. The parameter $\alpha$ is used only during decoding and is tuned over {0.1, 0.2} after selecting a suitable $\beta$, thus incurring minimal hyperparameter-tuning cost.
>
> Here, we mainly present a sensitivity analysis of the training-related hyperparameter $\beta$, while keeping $\alpha$ fixed. We evaluate $\beta$ over the range {0.1, 0.2, 0.3, 0.4, 0.5, 0.6}, and summarize the results in the following table.
>
> | Dataset | Baseline HR@10 (β=1.0) | Optimal β | Max HR@10 (within effective β) | Min HR@10 (within effective β) | Effective β Range (HR@10 & NDCG@10 > Baseline) |
> |:--------|:----------------------:|:---------:|:--------------------------:|:--------------------------:|:----------------------------------------------:|
> | CDs     |         0.1040         |    0.2    |      0.1077 (β=0.2)        |      0.1066 (β=0.4)        |                  [0.1, 0.6]                   |
> | Games   |         0.0773         |    0.2    |      0.0942 (β=0.2)        |      0.0777 (β=0.6)        |                [0.1, 0.4] $\cup$ [0.6]                |
> | Toys    |         0.1029         |    0.5    |      0.1063 (β=0.5)        |      0.1038 (β=0.2)        |                  [0.2, 0.6]                    |
> | Books   |         0.0315         |    0.1    |      0.0422 (β=0.1)        |      0.0335 (β=0.6)        |                  [0.1, 0.6]                    |
>
> As shown, our method is relatively less insensitive to the exact choice of $\beta$: it can achieves strong performance across a wide range of values in each dataset. However, the suitable range of $\beta$ may vary across datasets.

---

> ### Author Response · Authors · 2025-08-09
> **Less Than 10 Hours Remaining for Reviewer–Author Discussion, Please Join Discussion**
>
> Dear Reviewer,
>
> With less than 10 hours remaining in the reviewer–author discussion period, we sincerely request that you review our rebuttal and join the discussion. We have received feedback from all other reviewers, and they have confirmed that their concerns have been addressed. We believe your concerns have also been resolved.
>
> Best regards,
> Authors

---

### Author Response · Authors · 2025-08-06
**Request for AC to Prompt Reviewers for Their Feedback**

Dear Area Chair,

As the discussion phase is ending soon, we would greatly appreciate it if you could help prompt the reviewers to provide feedback.

We believe that we have carefully addressed their insightful comments in our response, but have not received any replies from three of the four reviewers. We’re concerned this may affect the outcome, and are happy to provide any further clarification if needed.

Thank you very much for your support!

Best regards,
The Authors

---

### Note · Authors · 2025-08-13

We sincerely thank the Area Chair and all reviewers for their time and insightful feedback (R1: v8sa; R2: QLsy; R3: 14G4; R4: YKBt). We are very pleased that reviewers found our information-theoretic approach to modeling token decisiveness to be novel, intuitive, and well-justified (R2, R3, R4). Reviewers also highlighted that our method is simple and practical to integrate into existing models (R1, R2), with promising results demonstrated through extensive experiments (R1, R2, R3, R4).

We are glad that we were able to thoroughly address the reviewers' concerns with additional experiments and clarifications. Specifically:

- In response to R1's concerns about our experimental setup, we clarify that it reflects real-world scenarios, follows widely adopted practices, and is free from data leakage. We also provide experiments to verify that 1) there are drifts between training and testing, 2) our method could still work under different data splitting.


- Regarding our binary weighting scheme (R1, R4), we provided an in-depth analysis and empirical results to demonstrate the superiority of our design.

- We also validated our method's generalizability on a new dataset (R2, R3), benchmarked against new baselines (R2), analyzed its computational cost (R3), and showed its positive impact on diversity (R3).

Though we unfortunately did not receive a response from R1, we believe we have addressed her/his concerns. Thanks to all the valuable feedback, we will supplement the final manuscript with these additions to make our paper more complete.

Finally, we express our great appreciation for the opportunity to strengthen our work. We are excited about our method's potential to enhance LLM-based recommenders and believe this paper offers a valuable contribution that will stimulate further discussion in the field.

---

### Decision · Program_Chairs · 2025-09-17

**Decision:**

Accept (poster)

**Comment:**

This paper proposes a simple yet effective method to reweight the tokens by information gain (over the item catalog distribution) to effectively improve training and generation for recommendation. This is motivated by the empirical observation that most tokens have low information gain but often correspond to high logits. Empirical studies demonstrate that the proposed weighting scheme helps with various existing LLM-based recommender systems.

The paper initially received negative-learning scores. During the rebuttal periods, the authors provided clarification as well as additional results with more baselines/datasets, which convinced the reviewers with negative scores to raise to borderline accept. Overall, all the reviewers agree that the proposed method is conceptually simple and intuitive and it can improve various methods with relatively small overhead. However, the negative concerns are around 1) the proposed reweighting scheme is more like a heuristic based on empirical observation and less theoretically motivated, 2) estimating item entropy from the training set can be potentially problematic, especially in a real-world setting where the item distribution is ever-changing. It is less clear how the proposed method can respond to such settings. I think despite the negatives, there are some interesting insights. However, I wouldn't be upset if this paper doesn't end up getting accepted due to the concerns mentioned above.